# Development of a fully automated chemiluminescent immunoassay for the quantitative and qualitative detection of antibodies against African swine fever virus p72

Lei Wang,[1,2] Duan Li,[1] Daoping Zeng,[3] Shuangyun Wang,[1] Jianwen Wu,[1] Yanlin Liu,[1] Guoliang Peng,[2] Zheng Xu,[1] Hong Jia,[4] Changxu Song[1]

**ABSTRACT** African swine fever (ASF), caused by ASF virus (ASFV), is a highly infectious and severe hemorrhagic disease of pigs that causes major economic losses. Currently, no commercial vaccine is available and prevention and control of ASF relies mainly on early diagnosis. Here, a novel automated double antigen sandwich chemiluminescent immunoassay (DAgS-aCLIA) was developed to detect antibodies against ASFV p72 (p72-Ab). For this purpose, recombinant p72 trimer was produced, coupled to magnetic particles as carriers and labeled with acridinium ester as a signal trace. Finally, p72-Ab can be sensitively and rapidly measured on an automated chemiluminescent instrument. For quantitative analysis, a calibration curve was established with a laudable linearity range of 0.21 to 212.0 ng/mL ($R^2$ = 0.9910) and a lower detection limit of 0.15 ng/mL. For qualitative analysis, a cut-off value was set at 1.50 ng/mL with a diagnostic sensitivity of 100.00% and specificity of 98.33%. Furthermore, antibody response to an ASF gene-deleted vaccine candidate can be accurately quantified using this DAgS-aCLIA, as evidenced by early seroconversion as early as 7 days post-immunization and high antibody levels. Compared with available enzyme-linked immunosorbent assays, this DAgS-aCLIA demonstrated a wider linearity range of 4 to 16-fold, and excellent analytical sensitivity and agreement of over 95.60%. In conclusion, our proposed DAgS-aCLIA would be an effective tool to support ASF epidemiological surveillance.

**IMPORTANCE** African swine fever virus (ASFV) is highly contagious in wild boar and domestic pigs. There is currently no vaccine available for ASF, so serological testing is an important diagnostic tool. Traditional enzyme-linked immunosorbent assays provide only qualitative results and are time and resource consuming. This study will develop an automated chemiluminescent immunoassay (CLIA) that can quantitatively and qualitatively detect antibodies to ASFV p72, greatly reducing detection time and labour-intensive operation, and improving detection sensitivity and linearity range. This novel CLIA would serve as a reliable and convenient tool for ASF pandemic surveillance and vaccine development.

**KEYWORDS** ASFV p72, chemiluminescent immunoassay, antibody detection

African swine fever (ASF), caused by ASFV virus (ASFV), is a highly contagious and acute hemorrhagic disease in domestic pigs and wild boars. ASF was first identified in Kenya in 1921 (1). Since 2007, ASF emerged outside Africa in Georgia and has spread to more than 60 countries and regions (2, 3). ASF entered China in August 2020 (4). ASF has caused huge losses to the global pig industries. Unfortunately, there are no vaccines or effective drugs to control ASF. Effective approaches to control and prevent ASF include early diagnosis (5).

Address correspondence to Changxu Song, cxsong2004@163.com, or Hong Jia, jiahong80@126.com.

Lei Wang and Duan Li contributed equally to this article. Author order was determined in order of increasing seniority.

The authors declare no conflicts of interest.

To ensure early diagnosis of ASF, many types of serological tests have been developed. Traditional enzyme-linked immunosorbent assay (ELISA) and lateral flow immunoassays (LFIA) are commercially available. New technologies such as luciferase immunosorbent assay (LISA) (6), nanoplasmonic biosensor (7), QDM-based ASFV immunosensor (QAIS) (8) and fluorescence immunochromatography test strip (FICTS) (9) have also been discussed. Serological tests are widely used for diagnosis of suspected infection, surveillance, epidemiological assessment and vaccine development and evaluation. Of these, ELISA is the most commonly used method. However, it is time-consuming, labour-intensive and has a narrow dynamic range (10). In addition, ASFV infection and effective immunity are closely related to antibody response (11). Quantification of antibody responses or conversion rates in vaccinated individuals can provide information not only for estimating vaccine responses and duration of protection, but also for improving vaccine immunogenicity, dosage optimization, amount, and time intervals (12). The aim of this study is to develop a novel method for the measurement of antibodies against ASFV that could provide high throughput, automation, and quantification.

ASFV is a large double-stranded DNA virus of the genus *Asfivirus* in the family *Asfarviridae* (13). The ASFV genome contains more than 150 open reading frames (ORFs) (14). The major capsid protein (MCP) p72, encoded by the B646L gene, is the main structural protein and constitutes approximately ~31%–33% of the total virion mass (15). Previous studies have shown that p72 can mediate viral entry by binding to the host factor, CD1d, and antibodies to p72 (p72-Ab) have significantly affected ASFV infection (16, 17). Therefore, p72-Ab is an important indicator of infection or herd immunity. Many serological tests based on p72 have been developed, such as colloidal gold strip and various types of ELISA, which have shown high reliability and sensitivity (18–21). However, there is no reliable automated quantitative test for p72-Ab.

Chemiluminescence immunoassay (CLIA) is a type of labeled immunoassay that combines a chemiluminescence system with an immune reaction. Based on the principle of light emission reactions, it is divided into direct chemiluminescence (CLIA), chemiluminescent enzyme immunoassay (CLEIA) and electrochemiluminescence immunoassay (ECLIA) (22). Chemiluminescent reactions are usually divided into two types: flash type represented by acridinium ester (AE)-$H_2O_2$ system, and glow type represented by alkaline phosphatase (AP)$-$1,2-dioxetane (AMPPD) system and the horseradish peroxidase (HRP)-$H_2O_2$-luminol system (23). At present, CLIAs are widely used in various fields of life science, clinical diagnosis, food safety and environmental assessment, and drug analysis due to its significant advantages of high sensitivity, wide linear range, and automated detection (24).

In this study, a novel automated double antigen sandwich chemiluminescent immunoassay (DAgS-aCLIA) was established for the detection of p72-Ab in porcine serum. This assay used p72-coated magnetic particles (p72-MPs) as carriers and AE-labeled p72 (p72-AE) as a signal trace. By constructing a curve, p72-Ab can be measured quantitatively and qualitatively. Due to its high sensitivity, automation, and wider linearity range, this assay could be a promising option for ASF epidemiological surveillance.

## MATERIALS AND METHODS

### Serum samples

Standard serum samples positive and negative for ASFV were obtained from the National Center for Veterinary Culture Collection, China Veterinary Drug Administration. Serum from pigs immunized with ASFV gene-deleted vaccine candidate was provided by the program team of African swine fever gene-deleted vaccine research and development of the National Key R&D Plan program of China. A total of 83 samples including 60 negative samples obtained before 2017 and 23 positive samples were collected and confirmed using commercial kits, 159 clinical samples were obtained from pigs with obvious clinical

syndrome were obtained. All sera were inactivated at 56°C for 30 min and stored at −80°C until use.

Serum samples positive for classical swine fever virus (CSFV), pseudorabies virus (PRV), porcine circovirus type 2 (PCV2), porcine reproductive and respiratory syndrome virus (PRRSV), and foot-and-mouth disease virus (FMDV) were confirmed using commercial ELISA kits, inactivated at 56°C for 30 min and stored at −80°C until use.

All treatments were strictly performed according to the operation of the World Organization for Animal Health (OIE).

## Experimental procedure

The principle of the developed DAgS-aCLIA is illustrated in Fig. 1A. The style of the diagram is a reference to the study by Liu et al. (25). Briefly, high capacity p72-MPs were used as a solid phase to capture antibodies in porcine serum and incubated with p72-AE as a signal trace. After separation and washing of the specific antigen-antibody complex. Substrate A (0.1 M HCl with 0.1% $H_2O_2$) was added and allowed to react for 1 min, then substrate B (0.25 M NaOH with 2% Triton X-100) was added. The RLU was recorded at 5 seconds.

Figure 1B shows the procedure of the assay using an automated instrument for incubation, washing and high-throughput luminescence reading. First, the detection-related reagents including p72-MPs, p72-AE, washing solution and substrate solution were loaded into an automated machine (SMART6500, KEYSMILE, Chongqing, China) for

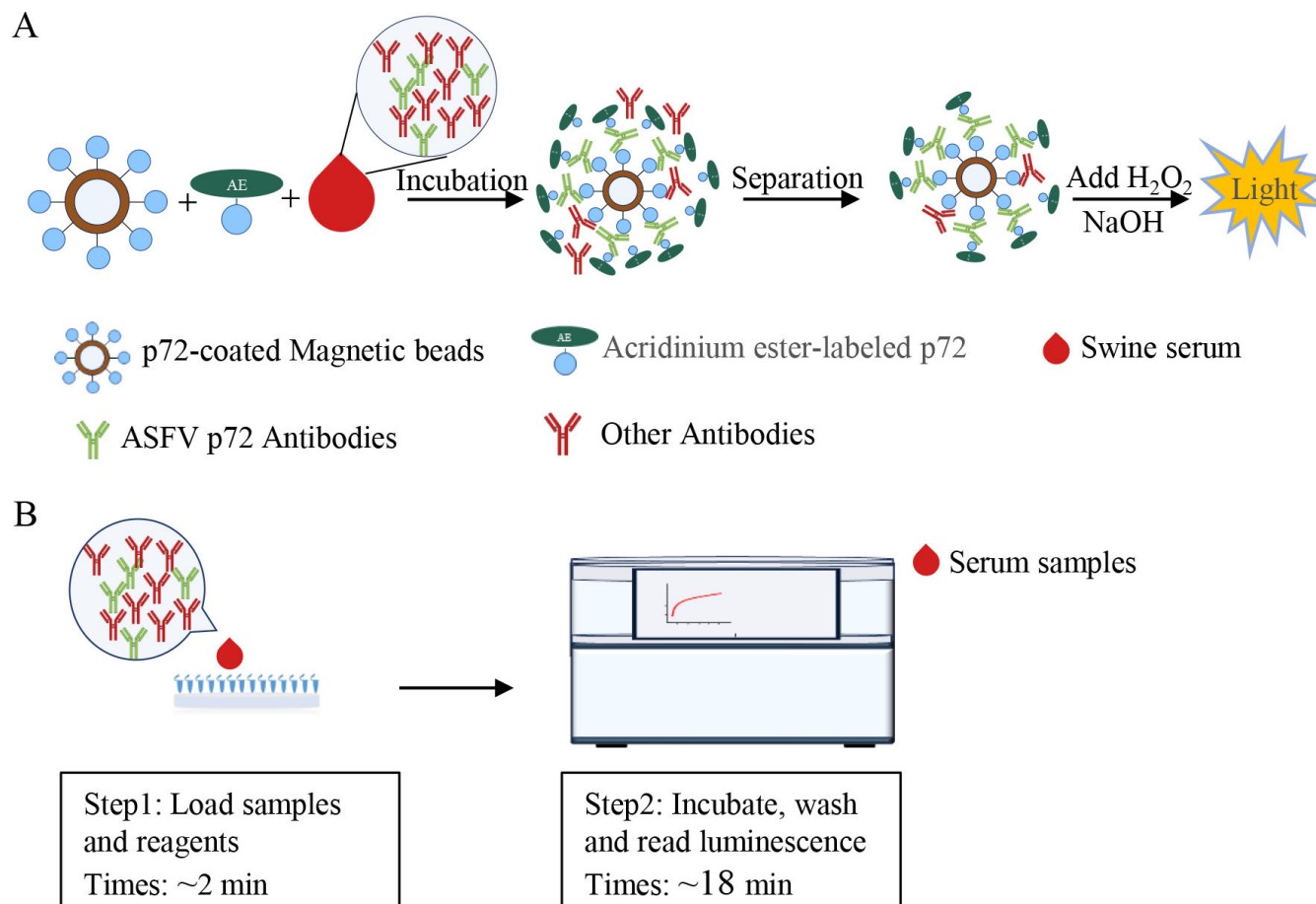

**FIG 1** Schemes of the principle and process of detecting ASFV p72 antibody by DAgS-aCLIA. (A) Schematic representation of p72-MPs and p72-AE to capture and detect p72-Ab. (B) Procedure for the determination of p72-Ab using the automated analyzer.

reaction. Then serum samples were loaded for a series of steps of incubation, washing and computation automatically. The concentration of ASFV p72-Ab can be produced.

## Expression and purification of p72 trimer

The ASFV genes *B646L* and *B602L* were separately codon optimized, synthesized and cloned into pCAGGS vector by Azenta (Suzhou, China). The constructed plasmids expressing p72 with His tag and p602L with Flag tag were co-transfected into CHO cells using TransIntroEL/PL transfection reagent (catalog no. FT231-02; TrasnGen Biotech, Beijing, China). The expression of recombinant p72 and pB602L was confirmed by immunoblotting (IB). The recombinant p72 was then purified by anti-His tag magnetic particles (catalog no. No70502; Beaver, Suzhou, China) and analyzed by sodium dodecyl sulfate-polyacrylamide gel electrophoresis (SDS-PAGE). The purified p72 was concentrated and applied to a Superdex 200 increase 10/300 GI size exclusion column (catalog no. 28–9909-44; GE Healthcare, Chicago, USA). The p72 from each elution was collected and imaged by negative-stain transmission electron microscopy (TEM) (26). Subsequently, a total of 20 particles were randomly selected and their diameters were determined using Image-Pro Plus 6.0 (Media Cybernetics, Inc., Rockville, MD, USA) by Servicebio. The purified p72 with nanoparticle size and morphology was collected for assays. The concentration of recombinant protein was measured using the enhanced BCA protein assay kit (catalog no. P0009; Beyotime, Shanghai, China).

## Conjugation of p72 to MPs

Purified p72 were coupled to MPs according to the recommended protocol for Tosyl beads. Briefly, 1 mg of MS160/Tosyl beads (catalog no. J-MS-S160T; JSR Life Sciences, Tokyo, Japan) were washed and resuspended in 900 µL binding buffer (0.1M HEPES buffer, pH 8.0), followed by addition of 50 µg purified p72, 500 µL catalytic reagent solution (1M sodium sulfate), and incubated for 18 h at 37℃ on a rotator. The reaction was then blocked with 15 µL blocking reagent (10% BSA) and incubated for a further,6 h at 37℃. After washing with washing buffer (TBS, pH7.4, 0.05% Tween20). The conjugation was characterized by scanning electron microscopy (SEM) and dynamic light scattering (DLS). Finally, the p72-MPs were dissolved in 1 mL phosphate buffer (pH 7.4, containing 150 mM NaCl, 1% BSA, 0.05% Tween-20, 0.05% ProClin300) and stored at 4℃ until needed.

## Conjugation of AE to p72

The purified p72 trimer was labeled with AE (NSP-SA-NHS; AE; catalog no.203126; MCT, Shenzhen, China) according to a previous study (27). Briefly, AE was dissolved in anhydrous dimethyl formamide (DMF) at 0.28 mg/mL. The purified p72 was diluted in labeling buffer (0.1M PBS, pH7.4, and 0.9% NaCl) at 0.16 mg/mL. Then, 60 µg p72 and 3 µg AE were added to 0.3 mL labeling buffer. The mixture was then incubated in the dark at 25℃ for 20 min with gentle shaking. The reaction was quenched by adding L-lysine at a final concentration of 0.1% for 30 min. The mixture was loaded onto a PBS-balanced G-25 desalting column and centrifuged at 1000 g for 2 min at 4℃. The eluent was collected in a brown bottle. This process was repeated for three times. with BSA and ProClin 300 were added to the eluent at concentrations of 1% and 0.05%, respectively, and stored at 4℃ until use. For long-term storage at −20℃, an equal volume of glycerol was added. Finally, 50 µL of the prepared p72-AE was added to a test tube and the relative light unit (RLU) was determined.

The labeling efficiency was defined as the ratio of molar concentration of AE to that of labeled protein. According to the manual, p72-AE was diluted with labeling buffer and its UV absorbance was measured at 280 nm. In our experiment, diluted p72-AE with an absorbance at 280 nm of 1.0 was selected and its pH was adjusted to 1.5 with hydrochloric acid. The UV absorbance of the adjusted p72-AE was measured at 367 nm. As a control, 1 mg/mL of p72 was diluted 50-fold with labeling buffer and its UV

absorbance was measured at 280 nm. The labeling efficiency was calculated using the following equation:$(1.0 - (A367 \times 0.17))/(A280 \times 50 \times M)$. M: molecular weight of p72.

## Determination of p72-Ab concentration in standard serum

50 µL of ASFV-positive standard serum was diluted with 550 µL of PBS in a 1.50 mL tube (Corning, Corning, USA), followed by addition of different volumes (10, 20, 50, 80 µL) of p72-MPs. The mixture was incubated for 2 h at room temperature (RT) on a rotary shaker. P72-Ab from ASFV negative serum and clinical positive serum were also extracted. A commercially available rabbit anti-p72 polyclonal antibody (catalog no. bs-41384R; Bioss, Beijing, China) was used as a control. The p72-Ab were separated from p72-MPs by using 40 µL of 0.1 M glycine in water (pH 2.5), followed by neutralization with 4 µL of 1 M Tris buffer (pH 9). Purified p72-Ab were analyzed by SDS-PAGE. The amount of p72-Ab was measured using an enhanced BCA protein assay kit. The concentration of p72-Ab in standard serum was calculated.

## Optimization of the reaction system

In order to obtain optimal conditions for DAgS-aCLIA, four parameters were optimized.

1.  Dilutions of p72-MPs. 100 µL of p72-MPs at dilutions of 1:20 1:30, 1:60, 1:100, and 1:200 was used to measure a standard positive sample (13.25 ng/mL) and a negative sample, the P/N ratio was calculated, where P is the RLU of the positive sample, N is the RLU of the negative sample.
2.  Dilutions of p72-AE. 100 µL of p72-AE at dilutions of 1:200, 1:500,1:1,000, 1:1,500, and 1:2,000 was used to measure ASFV positive (13.25 ng/mL) and a negative sample. The P/N ratio was calculated.
3.  Sample dilution buffers. Sample dilutions included normal swine serum (NSS), saline solution (SS, 0.9% sodium chloride), and PBS, and PBS containing 0.01% Tween 20, 1% casein, 15% horse serum (PBS-TCH) were tested. Additional preservative required when supplied in kit. Negative serum was tested 10 times and the influence of each dilution was assessed by the recovery of the values.
4.  Detection procedure. One-step: 100 µL p72-MPs, 20 µL serum sample, 100 µL p72-AE were added together in a test tube and incubated for 15 min at 37°C. After five washes with TBST (Tris buffered saline containing 0.1% Tween-20), 50 µL of substrate A and 50 µL of substrate B were added, the RLU was monitored. Two-step: 20 µL of serum sample was added to a test tube, then 100 µL of p72-MPs was added and incubated at 37°C for 15 min. After five washes with TBST, 100 µL p72-AE was added and incubated for a further 15 min at 37°C. After five washes, 50 µL of substrate A and 50 µL of substrate B were added and the RLU was monitored. In this assay, ASFV positive standard samples at concentrations of 13.25, 26.5, 53.0, and 106.0 ng/mL were detected and the linear regression equation was calculated.

## Quantitative analysis

Three important parameters were determined to conduct quantitative analysis.

1.  Calibration curves and linearity range. Establishing a calibration curve is a critical step for accurate quantitative bioanalysis (28). In this study, diluted ASFV positive standard sera were used as standard samples. The concentration of p72-Ab in the standard serum was calibrated as described above. ASFV positive standard serum with p72-Ab concentrations of 0.83, 1.66, 3.31, 6.63, 13.25, 26.5, 53.0, 106.0, 212.0, 424.0, 848.0, 1060.0, and 2120.0 ng/mL were measured to construct a calibration curve. The correlation coefficient ($R^2$) between RLU values and different concen-

trations of p72-Ab was analyzed using a four-parameter logistic (4PL) model with analyte concentration as abscissa (x value) and the logarithm of the RLU as ordinate (y value) (29).

The sigmoidal curves were fitted to an equation:

$$y = A2 + (A1 - A2)/(1 + (x/x0)^p)$$

The linearity range was analyzed using an equation:

$$y = ax + b$$

2. Lower detection limit (LDL). The detection limit is classified by the US National Bureau of Standards into three categories: instrumental limit of detection (LOD), method detection limit (MDL), and sample detection limit (SDL). In this article, we used the MDL to assess the LDL, which is defined as the lowest concentration of analyte in the test sample that reflects the sensitivity of the developed method (30). Briefly, the sample dilution was determined 20 times, and the corresponding concentration was calculated according to the established standard curve. The mean (M) and standard deviation (SD) were calculated and the LDL was then calculated using the equation: LDL = M + 2 SD.

## Qualitative analysis

To determine the cut-off value, diagnostic sensitivity and specificity of the developed DAgS-aCLIA. A total of 83 samples including 60 negative serum samples and 23 positive serum samples from individual pigs, were tested by the developed DAgS-aCLIA. The cut-off value, diagnostic sensitivity and specificity were calculated by receiver operating curve (ROC) analysis using MedCalc software (31).

## Cross-reactivity detection

Cross-reactivity was assessed by detecting positive sera against several porcine viruses, including CSFV, PRV, PCV2, PRRSV, and FMDV. ASFV positive and negative serum samples were used as controls. The concentrations of each sample were then calculated and compared.

## Reproducibility assessment

The reproducibility was evaluated according to a previous study with some modifications (32). Briefly, based on optimal conditions, different batches of p72-MPs and p72-AE were calibrated, one by one, by testing diluted ASFV positive and negative standard serum samples. One batch of p72-MPs and p72-AE was then introduced to test reproducibility. Eight serum samples (three negative control, three intermediate-positive control, and two strong positive control) were randomly selected. Three replicates of each sample were assayed in one batch to evaluate intra-assay variation and in separate batches to evaluate inter-assay variation. Mean (M), standard deviation (SD), and percent coefficient of variation (CV) were calculated.

## Stability analysis

All components including p72-MPs, p72-AE, substrate solution, sample dilution buffer and washing buffer were stored at 4°C for 0, 1, 2, 3, and 6 months, respectively. One negative serum and one positive serum were dispensed, and stored at −80°C until used.

## Detection antibody in ASFV- immunized pig sera

Serum of pigs immunized with an ASF gene-deleted vaccine candidate at different time points (0, 7, 14, 21, and 28 dpi; $n$ = 5/time point) was diluted 1:50. All diluted serum samples were tested by the established DAgS-aCLIA. The concentrations of each sample were then calculated based on standard curve.

## Comparison of analytical sensitivity and coincidence rate

Three commercial ELISA kits, including a blocking ELISA for p72-Ab (lote.210099; Ingenasa, Madrid, Spain), a competitive ELISA for ASFV p32 antibodies (lote. G79; ID.vet, Grabels, France) and an indirect ELISA (catalog no. Y.ME06C; Pu-tai, Luoyang, China) were used (5). ASFV positive standard serum and two clinical sera were diluted and detected to compare the detection procedure, analytical sensitivity, and linearity range between CLIA and ELISA, and 159 clinical samples were detected to compare the coincidence rate.

## Statistical analysis

All values had been maintained to 1–2 decimal places. Data analysis was performed using excel, OriginPro 7.0 (Microcal Software Inc., Northhampton, MA, USA), and GraphPad Prism version 8.0 software (San Diego, CA, USA). All data are presented as M ± SD. Statistical significance was set at $P < 0.05$.

## RESULTS

### Expression and purification of p72

In order to obtain recombinant p72 with high quality, ASFV p72 and pB602L were co-expressed in CHO cells. Their expression was confirmed by IB. As shown in Fig. 2A, the molecular weight (MW) of p72 was between 70 and 100 kDa, and that of pB602L was about 63 kDa. The expressed p72 was then purified by beads and molecular sieve chromatography, and further analyzed by SDS-PAGE and TEM. G-250 staining showed a sharp band around 75 kDa with a purity of more than 95% (Fig. 2B), and TEM revealed the nanoparticle size (with diameter of 8.46{plus minus}0.69 nm) and morphology of p72 are consistent with published data (Fig. 2C). Based on these results, the prepared p72 would be suitable for assay.

### Preparation and characterization of p72-MPs

The morphological change of p72-MPs was revealed from images of SEM. As shown in Fig. 2D, all MPs are uniformly dispersed, the blank particles had a porous surface, whereas the particles of p72-MPs were coated with a polymer shell. In addition, the diameters of p72-MPs were slightly increased.

The hydrodynamic diameter and its distribution of p72-MPs were measured by DLS. As shown in Fig. 2E, the diameters of the blank MPs and p72-MPs were approximately 1.76 and 2.25 µm, respectively. These results indicated that p72 was successfully immobilized on MPs.

### Preparation and characterization of p72-AE

The p72-AE was prepared and evaluated as described above, using multiple doses (1, 3, 9, 15, and 30 µg) of AE to label p72. According to Fig. 2F, the RLU increased with increasing doses of AE in the range of 1–30 µg. When the dose was higher than 9 µg, the increase started to slow down. In order to obtain the best luminescence performance and the most economical dose of AE without affecting the sensitivity of the experiment, the dose of 9 µg was used for followed assay.

To investigate the effect of the trimer of p72 on the luminescence activity, the nontrimeric forms of p72 were also labeled with AE and their chemiluminescence intensities and labeling efficiency were tested. It was found that the trimer of p72

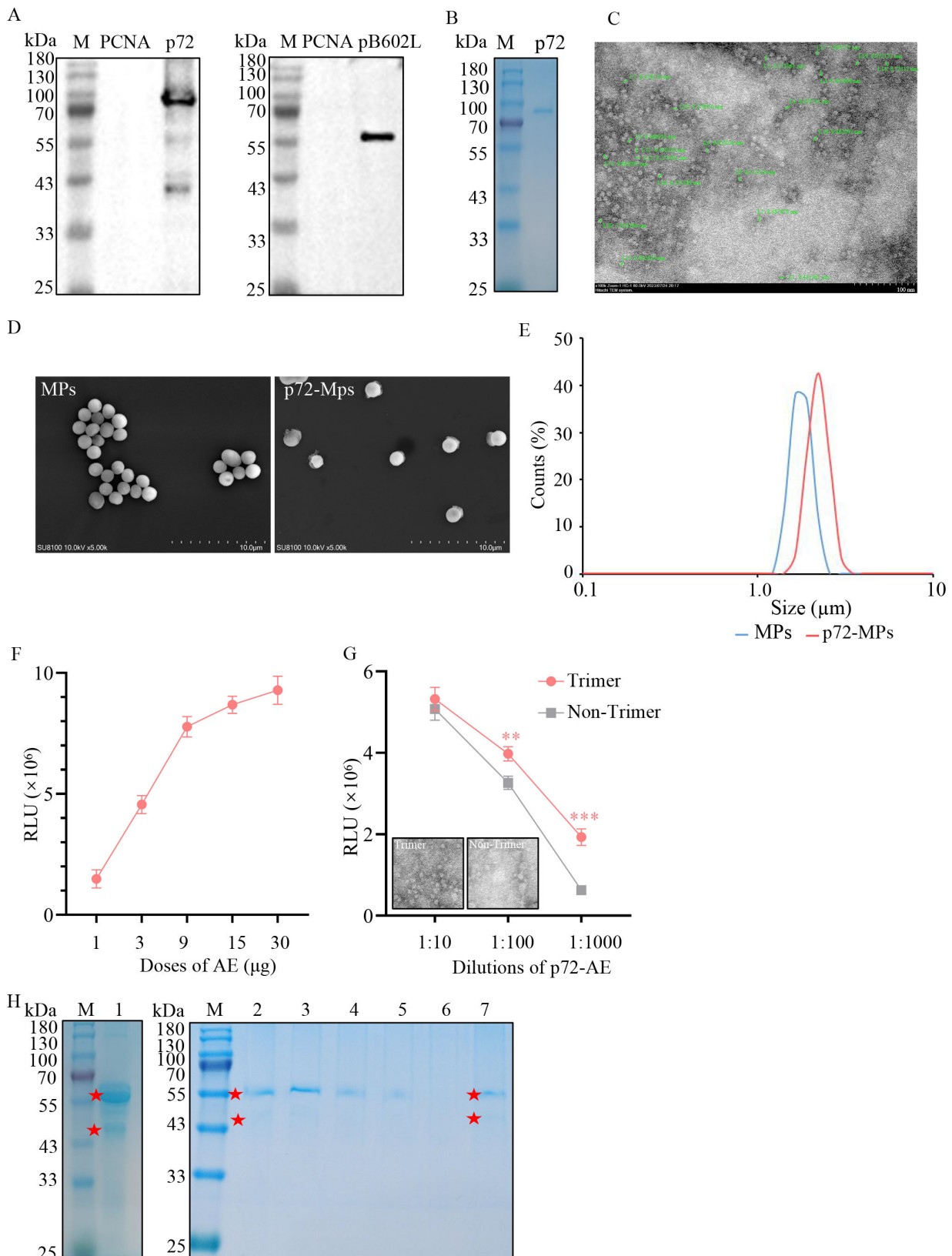

**FIG 2** Preparation of p72-MPs, p72-AE and determination of standard. (A) IB verification of the expression of p72 and its chaperone pB602L using anti-His tag and anti-Myc tag antibodies. (B) SDS-PAGE analysis of purified p72. (C) SEM images of purified p72. (D) SEM images of prepared MPs and p72-MPs. (E) Particle size analysis diagrams of prepared MPs and p72-MPs. (F) Effect of AE dose on chemiluminescent reaction, 60 µg of p72 was labeled with different doses (1, 3, 9, 15, (Continued on next page)

**Fig 2 (Continued)**

and 30 µg) of AE. (G) Effect of p72 conformation on chemiluminescent reaction, 60 µg of p72 with or without nano-size was labeled with 9 µg of AE. (H) Isolation of p72-Ab from ASFV positive standard serum. M: protein marker.1. commercial rabbit anti-p72 antibody; 2, 3, 4, and 5: positive standard serum, p72-MPs from 10 to 80 µL; 6 negative sera; 7: clinical positive serum. This assay was performed for three times. Major bands were marked with red pentagrams.

had higher chemiluminescence intensities as shown by a 3.08-fold increase (Fig. 2G). Subsequently, according to the calculation from three experiments, the labeling efficiency of p72 trimer and non-trimer was $6.27 \pm 0.08$ and $2.86 \pm 0.09$, respectively. These results demonstrated that p72-trimer has a more efficient use of AE, and the prepared p72-AE had excellent chemiluminescent activity, which has great potential for application in bioassays.

## Determination of p72-Ab concentration in ASFV positive standard serum

By antigen immunoaffinity chromatography purification, highly specific polyclonal antibodies can be isolated. As shown in Fig. 2H, the amount of p72-Ab isolated increased as increasing volume of p72-MPs. The p72-Ab can be only isolated from positive serum and showed two prominent bands consistent with a commercially rabbit anti-ASFV p72 antibody (catalog no.bs-41384; Bioss, Beijing, China). BCA analysis showed that the amount of p72-Ab in 50 µL of serum was approximately 1.06 µg. Therefore, the concentration of p72-Ab in ASFV positive standard serum was 21.20 µg/mL.

## Optimization of working conditions

In this study, the developed DAgS-aCLIA was applied to identify and quantify ASFV-Ab in pig serum. To achieve superior performance, crucial conditions were studied and optimized.

1.  Optimization of antigen concentration. Dilution ratios of p72-MPs and p72-AE were key parameters affecting immunoassay sensitivity and specificity. The P/N ratios increased rapidly with the dilution of p72-MPs and p72-AE until they reached a maximum. When the dilution of p72-MPs was higher than 1:60, the P/N ratio reached to the maximum value (Fig. 3A). When the dilution of p72-AE was 1:1000, the P/N ratio reached to the maximum value (Fig. 3B). Therefore, the dilution of p72-MPs at 1: 60 and p72-AE at 1:1,000 were used for further research.
2.  Optimization of sample diluent buffers. Four types of sample diluent buffers were tested, including NSS, SS, PBS, and PBS-TCH. The serum samples were diluted in these diluent buffers. As shown in Fig. 3C, sera diluted in PBS-TCH showed a significantly more stable values than sera diluted in other dilution buffers. The assay was performed in triplicate. Therefore, PBS-TCH buffer was selected for the subsequent tests.
3.  Optimization of detection procedure. In the experiment, one-step and two-step assays were performed under the same conditions. Positive sera with concentrations ranging from 13.25 to 106.0 ng/mL were detected. As shown in Fig. 3D, the $R^2$ of the one-step and two-step assays were 0.993 and 0.9923, respectively. This indicated that the two assays were all suitable for detection. The one-step assay was further selected due to a time saving of at least 15 minutes.

## Quantitative analytical performance

Under optimal experimental conditions, several parameters were determined, including calibration curve, linearity range and limit of detection.

1.  Calibration curve and range of linearity. A series of standard solutions (8.3–2,120.0 ng/mL) were prepared by diluting ASFV positive standard serum and

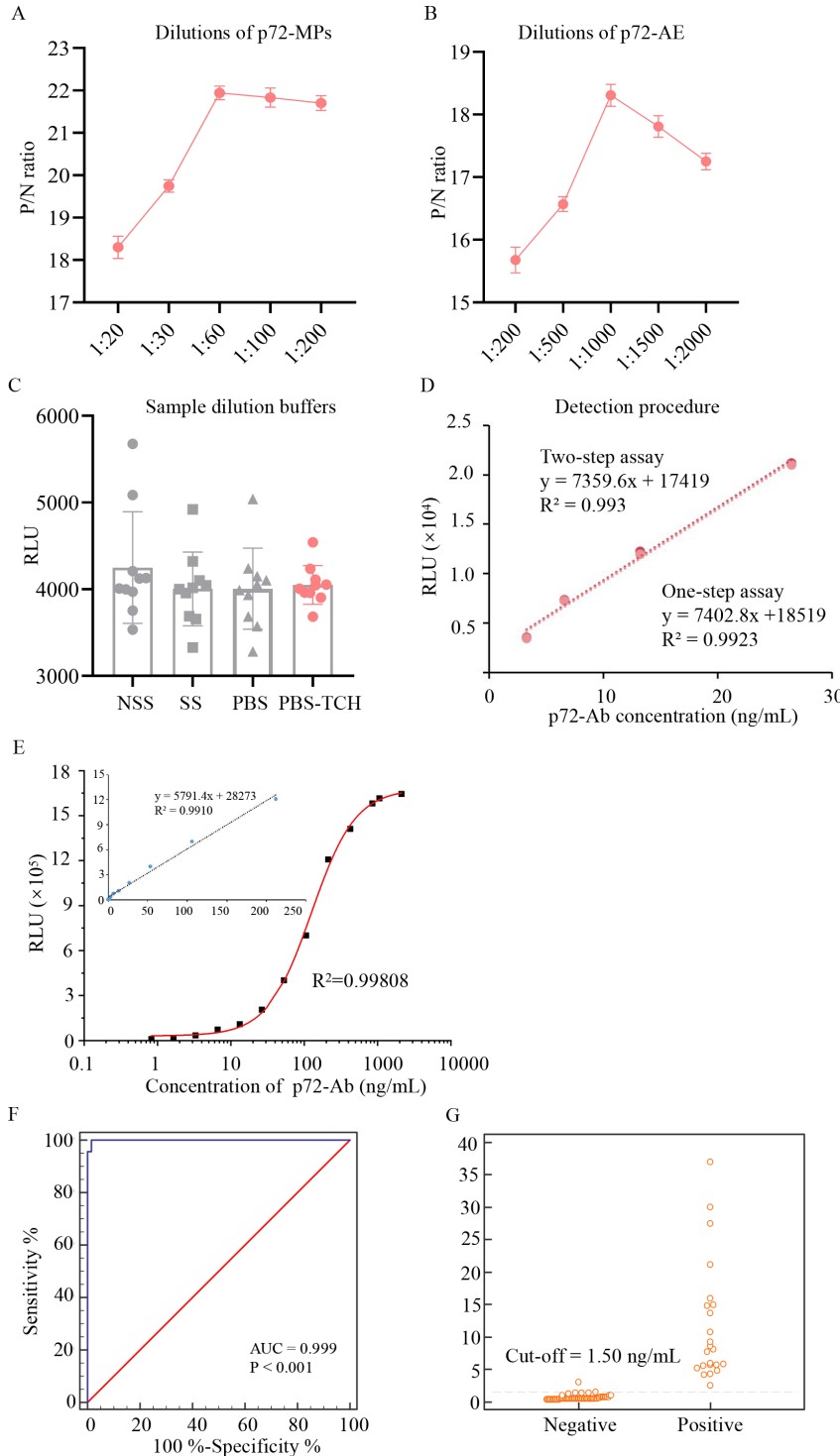

**FIG 3** Establishment of DAgS-aCLIA. (A) Optimization of p72-MPs, p72-MPs at dilutions of 1:20, 1:30, 1:60, 1:100, and 1:200 were evaluated in triplicate. (B) Optimization of p72-AE, p72-AE at dilutions of 1:200, 1:500, 1:1,000, 1:1,500, and 1:2,000 were evaluated in triplicate. (C) Optimization of sample dilution buffer. NSS, SS, PBS and PBS-TCH were used to dilute samples under same condition. (D) Detection procedure. One-step and two-step incubation of antigen with antibody were evaluated. (E) Establishment of a calibration curve for DAgS-aCLIA. The mean RLU in triplicate for standard samples with 0.83, 1.66, 3.31, 6.63, 13.25, 26.5, 53.0, 106.0, 212.0, 424.0, 848.0, 1060.0, and 2,120 ng/mL were measured under optimum conditions and plotted using Origin software. The logistic model was used for non-linear calibration. (F (Continued on next page)

Fig 3 (Continued)

and G) Determination of the cut-off value, each point on the ROC curve represents a sensitivity-specificity pair. AUC, area under the curve. Interactive dot diagram. 0, negative serum samples ($n = 60$); 1, positive serum samples ($n = 23$).

measured by DAgS-aCLIA. The calibration curve was shown in Fig. 3E. The RLUs increased with increasing concentration of p72-Ab. The RLUs of the standards ranged from 9,469.0 to 1644,272.0. The concentration of p72-Ab could be generated by an equation: $y = 1678{,}270.0 + (30{,}473.8–1678{,}270.0)/(1+(x/125.8)^{1.45})$, with $R^2 = 0.99808$, showing an excellent coefficient. For this study, we defined a correlation ($R^2$) greater than 0.9 as a "strong" correlation between two variables. As a result, p72-Ab can be determined with a linear quantification range of 0.21 to 212.0 ng/mL ($y = 5{,}791.4\,x + 28{,}273$, $R^2 = 0.9910$), and 0.21 to 424.0 ng/mL ($y = 3{,}686.2\,x + 87{,}736$, $R^2 = 0.9011$).

2. LDL. The limit of detection is an important performance parameter that is used both for characterizing the analytical sensitivity as well as interpreting the analysis results. In this study, the sample solution was detected to assess LDL. The LDL level was calculated to be around 0.15 ng/mL.

## Qualitative analytical performance

For qualitative analysis, a total of 83 pig serum samples were tested. The data were entered into the MedCalc software and the ROC curve analysis was performed to calculate the cut-off value and plot an interactive scatter plot. As a result, a diagnostic sensitivity of 100.00% and a diagnostic specificity of 98.33% were achieved, when the cut-off value was set at 1.50 ng/mL (Fig. 3F and G). The area under the curve (AUC) was 0.999, with $P < 0.0001$, as an AUC value greater than 0.9 indicates excellent diagnostic accuracy (33). These results demonstrate the excellent accuracy of the established DAgS-aCLIA.

## Cross-reactivity

The cross-reactivity of the DAgS-aCLIA was evaluated by positive and negative serum for ASFV, and positive serum for CSFV, PRV, PCV2, PRRSV, and FMDV. As shown in Fig. 4A, only the antibody concertation of the ASFV positive serum was above the cut-off, indicating that the proposed DAgS-aCLIA has no cross-reactivity with common viral pathogens and shows good specificity in practical applications.

## Repeatability of DAgS-aCLIA

Inter-assay and intra-batch reproducibility tests were performed (Fig. 4B). The intra-assay reproducibility test was performed by testing 8 samples using p72-MPs and p72-AE prepared in the same batch. The inter-assay reproducibility test consisted of testing 8 samples using p72-MPs and p72-AE prepared in different batches. The final results showed that the CV of the intra-batch assay was 0.69%–4.02% and the CV of the inter-batch assay was 0.71%–8.21%, indicating the high reproducibility of our method.

## Stability test

All components were stored at 4°C for 0–6 months. The same serum samples were tested at different months. As shown in Fig. 4C, the values of the positive samples were above the cut-off, and the negative controls were below the cut-off. The antibody concentration values remained stable, indicating that all detection reagents developed for ASFV p72-Ab have high stability during storage for 6 months.

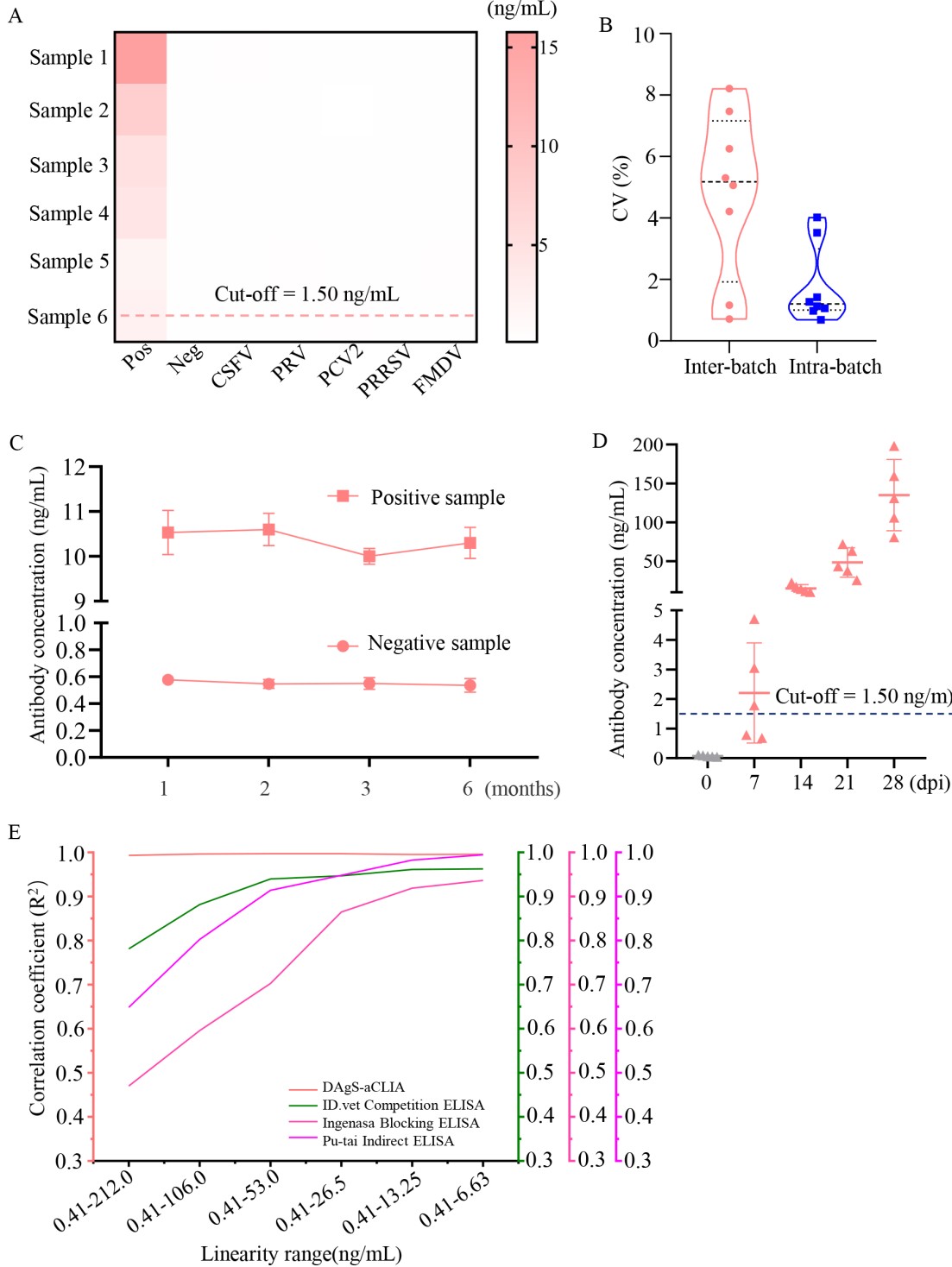

**FIG 4** Performance of DAgS-aCLIA. (A) Cross-reactivity. Sera positive for ASFV CSFV, PRV, PCV2, PRRSV, and FMDV, and negative for ASFV were tested. Data represent M from three independent experiments. (B) Results of repeatability test. The CVs of intra-batch and inter-batch were tested. (C) Stability of reagents. All reagents including p72-MPs, p72-AE, buffer and substrates were stored at 4°C for 1, 2, 3, and 6 months. One positive serum (>1.50 ng/mL) and one negative sample (<1.50 ng/mL) were detected. Data represent M and SD in triplicate. (D) Assessment of p72-Ab response to ASF gene-deleted vaccine candidate-immunized pigs. Serum samples were collected from pigs immunized with an ASF gene-deleted vaccine candidate (at 0, 7, 14, 21, and 28 dpi). (E) Analysis of linearity range of DAgS-aCLIA. Four types of ELISA including competition, blocking and indirect ELISA were compared by detecting standard serum with different ranges.

## Antibody response to p72 in ASFV gene-deleted vaccine candidate-immunized pigs

The antibody kinetics at different days in pigs immunized with an ASF gene-deleted vaccine candidate were evaluated. As shown in Fig. 4D, the concentration of p72-Ab in pigs gradually increased from 0 to 28 dpi. We found that the concentration of p72-Ab ranged from 0.68 to 3.06 ng/mL at 7 dpi, where the concentration of p72-Ab was detected above the cut-off (> 1.50 ng/mL) in three out of five pigs, and all seroconverted (>1.50 ng/mL) around 14 dpi, indicating a robust immune response in pigs elicited by this vaccine. These results demonstrate that DAgS-aCLIA has good potential for assessing immune efficacy.

## Comparative studies between CLIA and ELISA

To demonstrate the performance of DAgS-aCLIA in clinical practice, a comparative study was performed between the proposed CLIA, and commercial ELISA kits were performed. Any important parameters were evaluated.

1. Detection procedure. The proposed CLIA required only 20 minutes to automatically complete a sample test, including incubation, immunoreaction, washing, and result generation, whereas the traditional ELISA required more than 90 minutes of tedious operations. In particular, for the detection of immune complexes, the developed DAgS-aCLIA required only a few seconds after substrate addition, whereas the ELISA required at least 15 minutes for this step.
2. Analytical sensitivity. Serially diluted positive standard serum and clinical samples were selected for further testing. As shown in Table 1, clinical samples 1 and 2 were negative by ID.vet' and Ingenasa' ELISA but positive by CLIA and Pu-tai' ELISA. The ASFV positive standard serum diluted at 12800 was positive by CLIA, at 6400 by Pu-tai' ELISA, at 1600 by ID.vet' and at 800 by Ingenasa' ELISA. These data indicate that the developed DAgS-aCLIA has good sensitivity, significantly higher than that of ELISA.
3. Linearity range. The linearity range of CLIA and ELSIA at 0.41–212.0, 0.41–106.0, 0.41–53.0, 0.41–26.5, 0.41–13.25, 0.41–13.25, and 0.41–6.63 ng/mL were analyzed by Excel, respectively. The mean of $R^2$ in independent experiments was shown in Fig. 4E. All methods showed good linearity in three experiments. Of note, the developed DAgS-aCLIA had a wider range of linearity compared with the ELISA, ranging from 0.41 to 212.0 ng/mL, with $R^2$ of greater than 0.9. The linearity of the Pu-tai' and ID.vet' ELISAs was also good, ranging from 0.41 to 53.0 ng/mL ($R^2 >0.9$), whereas the linearity of the Ingenasa' ELISA was only 0.41 to 13.25 ng/mL. According to these results, the linearity range of the developed CLIA is 4 to 16 folds greater than that of the ELISA.
4. Coincidence rate. A total of 159 clinical samples were used for comparison. detected. As presented in Table 2, the DAgS-aCLIA assay showed 95.60%, 91.19%, and 94.97% concordance with commercial ELISA kits from Pu-tai, Ingenasa, and ID.vet, respectively. The concordance rates for positive samples were 89.19%, 72.97% and 83.78%. Of note, the concordance rates of negative samples between two methods were up to 97.54%, 96.72%, and 98.36%, indicating high specificity. In addition, the DAgS-aCLIA had the highest sensitivity in concordance with the above assay, as shown by a positivity rate of (37/159) 23.27%, which was higher than that of ELSIA. Consequently, the proposed DAgS-aCLIA is suitable for the detection of p72-Ab.

## DISCUSSION

ASF continues to spread worldwide. Nucleic acid detection is considered to be the most reliable method for early diagnosis of ASF. However, recent studies have shown that nucleic acids cannot be detected until more than 20 days after infection with naturally

**TABLE 1** The analytical sensitivity test of DAgS-aCLIA[a]

| Sample | Dilution ratios | DAgS-aCLIA | | ID.vet competition ELISA | | Ingenasa blocking ELISA | | Pu-tai ELISA | |
|---|---|---|---|---|---|---|---|---|---|
| | | AC | Result | S/N % | Result | X % | Result | S/P | Result |
| | 1:100 | 212.0 | + | 14.35 | | 98.44 | + | 3.05 | + |
| | 1:200 | 106.0 | + | 25.20 | + | 98.81 | + | 3.01 | + |
| | 1:400 | 53.00 | + | 39.98 | + | 89.96 | + | 2.70 | + |
| | 1:800 | 26.50 | + | 63.22 | − | 87.47 | + | 2.01 | + |
| ASFV standard | 1:1,600 | 13.25 | + | 81.61 | − | 66.08 | + | 1.45 | + |
| | 1:3,200 | 6.63 | + | 90.45 | − | 42.00 | Sus | 0.94 | + |
| | 1:6,400 | 3.31 | + | 100.64 | − | 23.13 | − | 0.58 | + |
| | 1:12,800 | 1.66 | + | 105.71 | − | 7.48 | − | 0.37 | − |
| | 1:25,600 | 0.83 | − | 108.10 | − | −4.70 | − | 0.15 | − |
| | 1:51,200 | 0.41 | − | 111.21 | − | −15.63 | − | 0.17 | − |
| | 1:2 | 3.72 | + | 65 | − | 34 | − | 0.93 | + |
| Clinical sample1 | 1:4 | 1.86 | + | 83 | − | 13 | − | 0.56 | + |
| | 1:8 | 0.93 | − | 88 | − | 2 | − | 0.35 | − |
| | 1:16 | 0.46 | − | 94 | − | −3 | − | 0.25 | − |
| | 1:2 | 3.16 | + | 41 | Sus | 1 | − | 0.57 | + |
| Clinical sample2 | 1:4 | 1.58 | + | 57 | − | 0 | − | 0.31 | − |
| | 1:8 | 0.79 | − | 80 | − | −69 | − | 0.17 | − |

[a]Sus: suspicious.

attenuated strains (34). Therefore, the detection of ASFV-Ab is valuable in the diagnosis of ASFV infection. In addition, great progress has been made in ASFV vaccine research, despite many challenges (35). Antibodies play a critical role in the protective immune response against ASFV (36). Large-scale assessment of prevalence and herd immunity to ASFV requires a high-throughput and quantitative screening. In this study, we established a novel DAgS-aCLIA for the rapid and automated quantification of p72-Ab in porcine serum.

CLIA uses chemiluminescent reactions in immunochemical assays and has been used for human and animal diseases in pandemics. Previous studies have shown that CLIA has great potential for the detection of antibodies against FMDV and CSFV, especially for FMDV, this CLIA can accurately differentiate virus-infected and vaccinated bovines (37–39), as well as antigen of PCV2 (40). For the serological determinants of coronavirus disease 2019 (COVID-19), canine parvovirus type 2 (CPV-2) and rabies virus (RABV), a fully automated CLIA demonstrated technological superiority in terms of labor and time-saving and high sensitivity (41–43).

The developed DAgS-aCLIA exhibited a remarkable specificity and sensitivity for ASFV-Ab (Fig. 4; Table 2). First, the advanced DAgS-aCLIA was based on the principle of double antigen sandwich (Fig. 1). It is capable of detecting IgM, IgA, and IgG (44, 45). As ASFV infection also has a polyclonal elevation of immunoglobulins including IgM,

**TABLE 2** Comparison of concordance rates between CLIA and ELISA[a]

| Methods | Samples | DAgS-aCLIA | | | |
|---|---|---|---|---|---|
| | | PS. N | NS. N | Total | Coincidence rate |
| | PS. N | 33 | 3 | 36 | 89.19% (33/37) |
| Pu-tai' ELISA | NS. N | 4 | 119 | 123 | 97.54% (119/122) |
| | Total | 37 | 122 | 159 | 95.60% ((33 + 119)/159) |
| | PS. N | 27 | 4 | 31 | 72.97% (27/37) |
| Ingenasa' ELISA | NS. N | 10 | 118 | 128 | 96.72% (118/122) |
| | Total | 37 | 122 | 159 | 91.19% ((27 + 118)/159) |
| | PS. N | 31 | 2 | 33 | 83.78% (31/37) |
| ID.vet' ELISA | NS. N | 6 | 120 | 126 | 98.36% (120/122) |
| | Total | 37 | 122 | 159 | 94.97% ((31 + 120)/159) |

[a]PS: positive serum; NS: negative serum; N, numbers.

IgA, and IgG. Our previous study also showed that a double antigen sandwich assay can provide improved sensitivity and specificity for ASFV-Ab detection (20). Second, the advanced DAgS-aCLIA used Tosyl MPs instead of a traditional 96-well plate (46). Although, 96-well plates have been extensively used in immunoassays. Its limited specific surface area affects antigen-antibody recognition efficiency, thereby reducing detection sensitivity (24). MPs have a larger specific surface area, super magnetism, and excellent biological compatibility (47). In addition, tosyl MPs can be easily manipulated compared with carboxyl MPs. MPs can enable automated detection, reducing manual operations and improving detection throughput and consistency.

The established CLIA was able to complete assay quickly, in less than 20 minutes, and was easy to operate with a large number of test samples. First, the advanced DAgS-aCLIA used AE as the signal probe. AE is a typical flash-type chemiluminescent substance (48). Signals can be recorded after 0.2 s (49). Whereas previous assays with AP or HRP in CLIA should have at least a few minutes for signal acquisition (50). AE has been widely used in the diagnosis of microbiology and infectious disease (51). Second, the advanced DAgS-aCLIA had an advantage of rapid automated detection. Using an automated analyzer, the computer can provide stable detection results. In addition to one-step sample addition and incubation, automated analysis was also available, detection time is greatly reduced. Compared with traditional ELISA, this method simplifies or replaces laborious processes such as long incubation, comparative analysis and repeated washing. When handling large numbers of samples, e.g., liquid transfer, clapper steps, the possibility of contamination has been effectively reduced. This simplified procedure has been applied to human T-cell lymphoma/leukemia virus type-1 (HTLV-1) (52). Take together, DAgS-aCLIA could overcome labor intensity and the need for professional skills.

The developed DAgS-aCLIA can provide quantitative and qualitative analysis of ASFV p72-Ab. Various assays have been designed to measure p72-Ab, but there are no reliable quantitative assays. In this study, p72-Ab in ASFV-positive standard serum was isolated through antigen affinity purification and used as a standard for absolute quantification of p72-Ab (Fig. 2). A wider range of linearity and LDL was obtained. The cut-off value was 1.50 ng/mL. The concordance rates with commercial ELISA kits were above 95% (Fig. 4). To control coronavirus disease 2019 (COVID-19), quantitative assays have been used (53, 54). In particular, the fully automated CLIA enables rapid and precise detection of COVID-19 antigen and antibody, thus playing an important role in COVID-19 diagnosis and vaccine development (55, 56).

To obtain best performance, we prepared p72 trimer with nano-size (Fig. 2). p72 is considered as a suitable candidate for ASFV serological test. In the ASFV capsid, p72 assembles into trimers (57). Consistent with previous studies, p72 trimers were produced in this study by co-expressing p72 and its chaperone pB602L in CHO. Compared with the non-trimer of p72, the trimer of p72 provided better luminescence properties (Fig. 2). Previous studies have shown that nano-sized platforms can efficiently immobilize AE molecules, which can facilitate the formation of OH and $O_2^-$ and lead to strong light emission (58).

## Prospects and conclusions

As errors cannot be avoided, we have a preliminary suggestion that international or national ASFV reference laboratories should be able to provide a common porcine p72-Ab standard to facilitate the calibration of all p72-Ab quantification assays, as in the case of COVID-19, etc. Moreover, it is possible to establish a multiplexed immunoassay to sequentially detect two different indicators in a single run, based on the flash and grow type of chemiluminescent reaction (59) or to quantify multiple antibodies using an automated chemiluminescent detection system (60). In addition, modern automated analyzers can support 360, 480, or even more tests per hour, enabling ultra-high throughput detection. These all help to improve test efficiency and is a unique advantage that ELISA methods do not have.

In conclusion, the developed DAgS-aCLIA has the advantage of high sensitivity, automation and quantitative detection of p72-Ab, which can provide important technological support for epidemiological surveillance and vaccine development.

## ACKNOWLEDGMENTS

We would like to thank the China Veterinary Drug Administration for providing the ASFV-positive and -negative serum, and the National Key R&D Plan program of China for providing the ASFV-immunized pig sera. We would like to thank Professors Hongfei Zhu and Hong Jia from the Institute of Animal Sciences, Chinese Academy of Agricultural Sciences, for providing help for serological experiments.

This work was supported by the National Key Research and Development Programs (2021YFD2001205) and Guangdong Basic and Applied Basic Research Foundation (2023A1515010234).

## AUTHOR AFFILIATIONS

[1]College of Animal Science, National Engineering Center for Swine Breeding Industry, State Key Laboratory of Swine and Poultry Breeding Industry, South China Agricultural University, Guangzhou, China
[2]Henry Fok School of Biology and Agriculture, Shaoguan University, Shaoguan, China
[3]Wens Foodstuff Group Co., Ltd, Yunfu, China
[4]Institute of Animal Sciences, Chinese Academy of Agricultural Sciences, Beijing, China

## AUTHOR ORCIDs

Lei Wang http://orcid.org/0000-0002-1359-3293
Hong Jia http://orcid.org/0000-0002-8062-1389
Changxu Song http://orcid.org/0000-0001-5311-0439

## ADDITIONAL FILES

The following material is available online.

### Open Peer Review

**PEER REVIEW HISTORY (review-history.pdf).** An accounting of the reviewer comments and feedback.

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
