## [Reviewer comments · Microbiology Spectrum]

Microbiology Spectrum

Development of a fully automated chemiluminescent immunoassay for the quantitative and qualitative detection of antibodies against African swine fever virus p72

Lei Wang, Duan Li, Daoping Zeng, Shuangyun Wang, Jianwen Wu, Yanling Liu, Guoliang Peng, Zheng Xu, Hong Jia, and Changxu Song

Corresponding Author(s): Lei Wang, South China Agricultural University College of Animal Science

Review Timeline:

Submission Date:	March 28, 2024
Editorial Decision:	April 26, 2024
Revision Received:	June 10, 2024
Editorial Decision:	June 23, 2024
Revision Received:	June 28, 2024
Accepted:	July 1, 2024

Editor: Vera Tesic

Reviewer(s): The reviewers have opted to remain anonymous.

Transaction Report:

DOI: <https://doi.org/10.1128/spectrum.00809-24>

Re: Spectrum00809-24 (**Development of a fully automated chemiluminescent immunoassay for the quantitative and qualitative detection of antibodies against African swine fever virus p72**)

Dear Dr. Lei Wang:

Thank you for the privilege of reviewing your work. Below you will find my comments, instructions from the Spectrum editorial office, and the reviewer comments.

Revision Guidelines

Sincerely,
Vera Tesic
Editor
Microbiology Spectrum

Reviewer #1 (Comments for the Author):

This study details the development of an automated double antigen sandwich chemiluminescent immunoassay for the detection of antibodies against ASFV p72. The immunoassay utilizes both purified recombinant p72 trimers coupled to magnetic particles and trimers labeled with the chemiluminescent reporter acridinium ester to form a sandwich complex when anti-p72 is present. Overall, the properties and performance of the assay is well-characterized and presented in a well-organized manner in the

manuscript.

The manuscript contains several grammatical errors that are somewhat distracting from an otherwise well-constructed manuscript; some examples of these errors are as follows:

Page 3, line 76: replace "serological testes" with "serological tests"

Page 3, lines 78-79: sentence is confusing, please rephrase

Page 4, line 94: "72" should be "p72"

Page 4, line 107: add "and" before "drug analysis"

Page 4, line 112: replace "signal trance" with "signal trace"

Page 5, line 140: replace "Triton-100" with "Triton X-100"

Page 6, line 167: replace "sulphate" with "sulfate"

Page 8, line 212: replace "stability" with "recovery"

Page 8, lines 214-215 and 217: what do "after five times" and "after three times" refer to? Addition of p72-AE? Please clarify.

Page 9, lines 247-248: "To determine the cut-off value, ..." is an incomplete sentence.

Page 10, line 290: replace "molecular sieve" with "molecular sieve chromatography"

Page 10, line 293: replace "in" with "is"

Page 12, line 335: should "1:10000" be "1:1000"?

Figure 3, panel D: graph contains a watermark that should be removed

Table 2: replace "coincidence" with "concordance"

Some additional questions and/or specific comments are as follows:

- Page 4, line 98: CLIA has been around proposed and utilized for over 40 years, so I would not characterize it as "new".
- Page 10, lines 292-293: size and morphology of p72 can be more clearly shown with scale bar and close-up of a few representative particles.
- Is the 3-fold increase in chemiluminescence intensity seen with the p72 trimer likely due to there being an AE label on each monomer? How reproducible is the AE labeling efficiency from batch to batch?
- What is the throughput of the automation described in this manuscript? Was sample carryover assessed in this automated test format?
- Were any of the ASFV positive and negative samples verified by an alternate, orthogonal method such as PCR or another molecular genetic technique?
- In Figure 4, panel A: were there 6 samples for each category and the signal was zero except for 6 sera positive for ASFV?
- How would the calibration from one batch/lot of reagents (p72-MPs and p72-AE) be assessed and how robust is it from batch/lot to batch/lot? Given the lack of a common porcine p72-Ab standard, can the authors save and retest a set of ASFV positive standard sera to determine the comparability of different reagents and calibrations based on the reproducibility of the set of measured concentrations?

Reviewer #2 (Comments for the Author):

In the manuscript "Development of a fully automated chemiluminescent 1 immunoassay for the quantitative and qualitative detection of antibodies against African swine fever virus p72" the authors Lei Wang et al., have developed a novel automated

double antigen sandwich
chemiluminescent immunoassay (DAGS-aCLIA) to detect antibodies against ASFV p72 (p72-Ab). The overall methods, results and discussion are good and convincing. Find detailed comments in the attachment.

In the manuscript “Development of a fully automated chemiluminescent 1 immunoassay for the quantitative and qualitative detection of antibodies against African swine fever virus p72” the authors Lei Wang et al., have developed a novel automated double antigen sandwich chemiluminescent immunoassay (DAgS-aCLIA) to detect antibodies against ASFV p72 (p72-Ab). The overall methods, results and discussion are good and convincing. However, the manuscript was poorly written and hard to understand the language or meaning of some sentences. Please review the language carefully and correct all the typos.

Major comments:

- 1) Please indicate whether any relevant IRB was required for the study and if required, received approval or exempt etc.,
- 2) Though the amount and dilutions of p72-MPs and p72-AE were mentioned, the amount of swine serum that was used for each assay seems not to be mentioned anywhere in the manuscript. Please consider adding the volumes of serum used for development of this assay.

All other comments:

Introduction

Line 71 “To date, many different serological assays have been established.” This statement needs a better phrasing to make it sound logical

Line 74 Expand QDM in QDM-based ASFV immunosensor

Line 76 been discussed. Serological testes are

Line 78 Among these, ELISA as most commonly used method, its time-consuming and labor79

Results

Figure4

Line749 The CVs of inter-batch and inter-batch were tested” should be intra and inter

750 reagents including p72-MBs = should be p72-MPs

The Figure 4 sections has E where as the legend has G, without E and F, please clarify

Discussion

The DAgS-aCLIA is capable of detecting IgM, IgA and IgG- should be clarified in which infection/detection situations to the appropriate references 44,45

It’s not clear in the below statement what the exact differentiation factor is for tosyl MPs and how they are done, ex: sample cup or another reaction chamber vs 96-well plate

“ Second, the advanced DAgS-aCLIA used Tosyl MPs instead of a traditional 96-well plate”

Prospects and conclusions

The ending “also has very good prospects for application.” Should have a conclusive point instead of just application.

Reviewer #1:

This study details the development of an automated double antigen sandwich chemiluminescent immunoassay for the detection of antibodies against ASFV p72. The immunoassay utilizes both purified recombinant p72 trimers coupled to magnetic particles and trimers labeled with the chemiluminescent reporter acridinium ester to form a sandwich complex when anti-p72 is present. Overall, the properties and performance of the assay is well-characterized and presented in a well-organized manner in the manuscript.

The manuscript contains several grammatical errors that are somewhat distracting from an otherwise well-constructed manuscript; some examples of these errors are as follows:

Dear Reviewer:

Thank you very much for taking your time to review our manuscript. All of us have carefully studied your suggestions and comments, and agreed that these suggestions and comments are very helpful in revising and improving our manuscript. We really appreciate all your generous suggestions and comments. We have carefully made revisions that we hope will meet with your approval.

1. Page 3, line 76: replace "serological testes" with "serological tests"

Response: Thank you for your careful review and kind suggestions. We sincerely apologize for the language problems in the original manuscript. And we have corrected the spelling error above .

2. Page 3, lines 78-79: sentence is confusing, please rephrase

Response: Thanks for your reasonable suggestion! We have made revision and included the sentences "Of these, ELISA is the most commonly used method. However, it is time-consuming and labour-intensive and has a narrow dynamic range.". We hope that it is now clearer.

3. Page 4, line 94: "72" should be "p72"

Response: We are very sorry for our careless mistakes. We have corrected this error.

4. Page 4, line 107: add "and" before "drug analysis"

Response: Thanks for your nice suggestion! We have added "and" before "drug analysis".

5. Page 4, line 112: replace "signal trance" with "signal trace"

Response: We apologize for the mistake in the original manuscript. We have corrected it in the current version of the manuscript.

6. Page 5, line 140: replace "Triton-100" with "Triton X-100"

Response: Thank you for your careful review. We have corrected "Triton-100" to "Triton X-100" in the current version of the manuscript.

7. Page 6, line 167: replace "sulphate" with "sulfate"

Response: Thank you for your kind suggestion. We have changed "sulphate" to "sulfate" in the current version of the manuscript.

8. Page 8, line 212: replace "stability" with "recovery"

Response: We are grateful for the suggestion! We have replaced "stability" with "recovery" in the current version of the manuscript.

9. Page 8, lines 214-215 and 217: what do "after five times" and "after three times" refer to? Addition of p72-AE? Please clarify.

Response: We feel very sorry that we didn't express our self clearly. In the manuscript, "after five times" and "after three times" refer to "after five washes with TBST" and "after three washes with TBST", respectively. We have included the sentences "After five washes with TBST (Tris buffered saline containing 0.1% Tween-20)" and "After three washes with TBST" in lines 226 and 229 of the current version of the

manuscript. However, after careful checking, we have not found "Addition of p72-AE" in line 217 of the original manuscript. If there's anything else that needs fixing, please let us know! Your comments are greatly appreciated.

10. Page 9, lines 247-248: "To determine the cut-off value, ..." is an incomplete sentence.

Response: Thanks for the kind and insightful comments. We have included the sentences "To determine the cut-off value, diagnostic sensitivity and specificity of the developed DAgS-aCLIA".

11. Page 10, line 290: replace "molecular sieve" with "molecular sieve chromatography"

Response: We are grateful for the suggestion! We have replaced "molecular sieve" with "molecular sieve chromatography".

12. Page 10, line 293: replace "in" with "is"

Response: Thank you for your kind suggestion. We have replaced "in" with "is" in the current version of the manuscript.

13. Page 12, line 335: should "1:10000" be "1:1000"?

Response: We sincerely thank you for careful reading. we have corrected "1:10000" to "1:1000".

14. Figure 3, panel D: graph contains a watermark that should be removed

Response: We are very sorry for our careless mistakes. We have removed this watermark shown in the revised figure 3 D.

15. Table 2: replace "coincidence" with "concordance"

Response: Thank you for your kind suggestion! Exactly, a more appropriate word is "concordance". We have replaced "coincidence" with "concordance", please see Table

2.

Some additional questions and/or specific comments are as follows:

1. *Page 4, line 98: CLIA has been around proposed and utilized for over 40 years, so I would not characterize it as "new".*

Response: Thank you for the insight on this matter. "New" is indeed a poor description. CLIA has been in development for more than 40 years, it is not a new technology, but it is gradually gaining attention in the field of veterinary medicine. In the manuscript, we have removed "new" to more accurately describe this technology.

2. *Page 10, lines 292-293: size and morphology of p72 can be more clearly shown with scale bar and close-up of a few representative particles.*

Response: We appreciate your valuable perspective. According to previous studies by Tsinghua University and China Agricultural University (Refs A and B), transmission electron microscopy (TEM) was used to image the p72 trimer and revealed a nano-sized p72. Therefore, in our study, TEM was also used to image p72 and, as expected, the size and morphology of the prepared p72 are consistent with the published data, as described in the manuscript. The scale bar has been added.

We all agree that the size and morphology of p72 can be shown more clearly with a close-up of a few representative particles. In fact, before resubmitting our manuscript, we have tried our best to image p72 with a few representative particles at high resolution using TEM. However, we didn't get the expected results, due to the limited resolution of the TEM, as shown in the following figure (Fig.A). We sincerely apologize for this.

Fig. A A few representative particles by TEM

Thank you again for your valuable suggestion. Following your suggestion, we performed additional analysis by randomly selecting 20 particles and measuring their

diameters using Image-pro plus 6.0 (Media Cybernetics, Inc., Rockville, MD, USA), and as shown in the following figure and table, and the size of p72 is 7-10 nm in diameter, and the average diameter of the particles is 8.46 ± 0.69 , which is consistent with the published data by Liu.,et al (Ref.C).

Finally, we included the sentences "Subsequently, a total of 20 particles were randomly selected and their diameters were determined using Image-pro plus 6.0 (Media Cybernetics, Inc., Rockville, MD, USA) by Servicebio" in lines 160-162 of the current version of the manuscript, and "TEM revealed the nanoparticle size and morphology of p72 with a diameter of 8.46 ± 0.69 nm, is consistent with published data (Fig. 2C)." in lines 308-310 of the current version of the manuscript. We also revised figure 2 C.

Figure	Diameter (nm)					
	1	2	3	4	5	Average
Img0635	8.431361	8.487877	8.42974	9.402607	8.27106	8.60
	8.652055	8.42812	8.46693	8.114328	7.028186	8.14
	8.43622	8.27106	9.862092	8.211388	8.80232	8.72
	8.727511	7.099736	9.862092	8.072132	8.17805	8.39
	Average	8.4617		SD	0.6868	

Ref A: Geng Menga., et al. Expression and purification of African Swine Fever virus p72 trimers as subunit vaccine candidate.bioRxiv.2020.

Ref. B: Geng Menga., et al.Structural design and assessing of recombinantly expressed African swine fever virus p72 trimer in *Saccharomyces cerevisiae*. Frontiers in Microbiology.2022.

Ref. C: Liu Q., et al. Structure of the African swine fever virus major capsid protein p72. Cell Research. 2019.

3. Is the 3-fold increase in chemiluminescence intensity seen with the p72 trimer likely due to there being an AE label on each monomer? How reproducible is the AE

labeling efficiency from batch to batch?

Response: In this study, the ratios of p72 and AE were investigated as described in the manuscript. Finally, 9 μg AE were used to label 60 μg p72, where the best luminescence performance and the most economical dose of AE can be obtained.

For AE labeling, its labeling efficiency can be defined as the ratio of molar concentration of AE to that of labeled protein in the mixture according to the instruction manual and operation manual. Briefly, the prepared p72-AE was diluted using labeling buffer and its UV absorbance was measured at 280 nm. Then, 100 μL of diluted p72-AE with Abs₂₈₀ of 1.0 was taken and its pH was adjusted to 1~2 with hydrochloric acid. The UV absorbance of the adjusted p72-AE at pH of 1.5 was measured at 367 nm. Then, 1mg/mL of p72 (unlabeled protein, defined as standard) was diluted using labeling buffer at 50-fold and its UV absorbance was measured at 280 nm. The labeling efficiency can be calculated using equation: $(1.0 - (\text{Abs}_{367} \times 0.17)) / (\text{Abs}_{280}(\text{standard}) \times 50 \times M)$. M: molecular weight of p72. After calculation, the labeling efficiency for p72 trimer and non-trimer were 6.27 ± 0.08 and 2.86 ± 0.09 , respectively.

According to a previous study, nanomaterial-based platforms could efficiently immobilize AE molecules and facilitate the formation of $\text{OH}\cdot$ and $\text{O}_2\cdot^-$, leading to strong light emission (Ref 58 in the original manuscript). Our results showed that p72 trimer may serve as a nanoplatform that can efficiently immobilize AE and facilitate luminescence reactions, due to its proper spatial conformation.

However, the ratio of the labeling efficiency of p72 trimer and non-trimer was about 2.19, which is less than 3.08. We speculated that a nano-sized p72 might have other advantages in facilitating luminescence reactions. For example, a proper spatial conformation of p72 allows more AE exposure to participate in luminescence reactions. Perhaps, it is like magnetic particles (MPs), which can efficiently display antigen, and increase the antigen contact area, thereby promoting the antigen-antibody recognition reaction, when compared to 96-well plates. A nano-sized p72 may have similar effect in facilitating luminescence reactions.

In the reproducibility experiments, we tested different batches of p72-AE and

p72-MPs, as shown in lines 387-393, the inter-batch assay was 0.71-8.21%, indicating the high reproducibility of our method. In this assay, we prepared different batches of p72-AE according to our method and found that the chemiluminescence intensity of different batches was similar.

Finally, we included the sentences "The labeling efficiency was defined as the ratio of molar concentration of AE to that of labeled protein. According to the manual, p72-AE was diluted with labeling buffer and its UV absorbance was measured at 280 nm. In our experiment, diluted p72-AE with an absorbance at 280 nm of 1.0 was selected and its pH was adjusted to 1.5 with hydrochloric acid. The UV absorbance of the adjusted p72-AE was measured at 367 nm. As a control, 1 mg/mL of p72 was diluted 50-fold with labeling buffer and its UV absorbance was measured at 280 nm. The labeling efficiency was calculated using the following equation: $(1.0 - (A_{367} \times 0.17)) / (A_{280} \times 50 \times M)$. M: molecular weight of p72" in lines 190-198 of the current version of the manuscript.

We included the sentences "To investigate the effect of the trimer of p72 on the luminescence activity. The non-trimeric forms of p72 were also labeled with AE and their chemiluminescence intensities and labeling efficiency were tested. It was found that the trimer of p72 had higher chemiluminescence intensities as shown by a 3.08-fold increase (Fig.2 G). Subsequently, according to the calculation from three experiments, the labeling efficiency of p72 trimer and non-trimer was 6.27 ± 0.08 and 2.86 ± 0.09 , respectively. These results demonstrated that p72-trimer has a more efficient use of AE, and the prepared p72-AE had excellent chemiluminescent activity, which has great potential for application in bioassays." in lines 328-336 of the current version of the manuscript.

Ref A: Zhili Han., et al. Acridinium ester-functionalized carbon nanomaterials: general synthesis strategy and outstanding chemiluminescence. ACS Appl. Mater. Interfaces. 2016

4. *What is the throughput of the automation described in this manuscript? Was sample*

carryover assessed in this automated test format?

Response: The instrument can be loaded with up to 16 sample holders carrying a total of 96 samples. Testing can be performed as soon as sampling is complete. Samples, reagents and reaction cups can be added at any time during the test, and approximately 300 samples (the instrument manual indicates 360 tests) can be tested in 1 hour using one-step model.

The instrument was installed with a series of calibration tests. In this study, the sample carryover rate was tested according to the instrument manual. It is important to note that the concentration of the high concentration sample is at least 10^5 times higher than the zero concentration sample. In brief, two samples were measured at high and zero concentrations. Each was repeated three times as a group, giving a total of five groups of measurements, and the carryover rate in each group was calculated using the formula: $K = (A4-A6)/(A-A6)$, where A4 is the measured value of the fourth sample in each group, while A6 is the measured value of the sixth samples in each group, A is the measured value of the original sample. The final mean of the measured K was less than 10^{-5} , which satisfies the requirement.

Samples	A	A1	A2	A3	A4	A5	A6
RLU	2,920,519	2,882,625	2,911,258	2,992,468	3911	4001	3925
Acceptance Value	$K \leq 10^{-5}$			K value		4.8^{-6}	

5. *Were any of the ASFV positive and negative samples verified by an alternate, orthogonal method such as PCR or another molecular genetic technique?*

Response: Thank you for your question. In this study, ASFV positive and negative standard sera were obtained from the National Centre for Veterinary Culture Collection (<https://syjg.agri.cn/sso-client-oauth2/shopping/index#/shopping>). Clinical samples were collected, inactivated and confirmed using commercial ELISA kits recommended by the OIE. There are no ASFV positive and negative samples verified by an alternative orthogonal method such as PCR or other molecular genetic techniques.

6. *In Figure 4, panel A: were there 6 samples for each category and the signal was zero except for 6 sera positive for ASFV?*

Response: Thank you. There are six samples for each pathogen. The signals for the other pathogens were not zero. The antibody concentration values for the other pathogens are well below the cut-off value.

7. *How would the calibration from one batch/lot of reagents (p72-MPs and p72-AE) be assessed and how robust is it from batch/lot to batch/lot? Given the lack of a common porcine p72-Ab standard, can the authors save and retest a set of ASFV positive standard sera to determine the comparability of different reagents and calibrations based on the reproducibility of the set of measured concentrations?*

Response: Thank you for your question. The established CLIA used p72-MPs as carrier and p72-AE as signal trace. Based on optimal conditions, different batches of p72-MPs and p72-AE were calibrated, one by one, by testing diluted ASFV positive and negative standard serum samples. We found that different batches of p72-MPs and p72-AE had good reproducibility. One batch of p72-MPs and p72-AE was then used in the developed CLIA. Therefore, the reproducibility of the developed CLIA was found to be good reproducibility as described.

We did not make this clear, and we all agree that your questions are very helpful in improving our manuscript. Therefore, we included the sentences "Based on optimal conditions, different batches of p72-MPs and p72-AE were calibrated, one by one, by testing diluted ASFV positive and negative standard serum samples. One batch of p72-MPs and p72-AE was then introduced to test reproducibility." in lines 270-273 of the current version of the manuscript.

Thank you for your comment, p72-Ab was isolated in different batches of ASFV positive standard serum samples and their levels were re-determined (Fig.A). Combined with our previous study shown in Figure 2 H, we found that there were only small differences in the antibody concentrations in different ASFV positive

standard serum samples. However, this does not affect our experiment, because in our experiment, specific batch of ASFV positive standard serum was used as reference to calibrate the assay. In the assay, specific concentrations such as 6.25, 12.5, 25 ng/mL) of the dilutions were used as standards only when they reached specified RLU values. The developed CLIA was then calibrated by determining these standard dilutions. Under these conditions, the developed CLIA has good reproducibility for clinical use. However, a common porcine p72-Ab standard is urgently needed to facilitate the calibration of all p72-Ab quantification assays, as this assay will become increasingly important for ASF epidemiological surveillance and vaccine development.

Fig.A Isolation of p72-Ab from different batches of ASFV positive standard sera.

R1-3: batches of positive sera.

Reviewer #2:

In the manuscript "Development of a fully automated chemiluminescent 1 immunoassay for the quantitative and qualitative detection of antibodies against African swine fever virus p72" the authors Lei Wang et al., have developed a novel automated double antigen sandwich chemiluminescent immunoassay (DAgS-aCLIA) to detect antibodies against ASFV p72 (p72-Ab). The overall methods, results and discussion are good and convincing. However, the manuscript was poorly written and hard to understand the language or meaning of some sentences. Please review the language carefully and correct all the typos.

Dear Reviewer:

Thank you very much for taking your time to review our manuscript. We really appreciate all your generous suggestions and comments. All of us have carefully studied your suggestions and comments, and agreed that these suggestions and comments are very helpful in revising and improving our manuscript. We have carefully made revisions that we hope will meet with your approval.

Major comments:

1) Please indicate whether any relevant IRB was required for the study and if required, received approval or exempt etc.,

Response: Dear Reviewer, Does "IRB" mean "Institutional Review Board Statement"? If so, our study is not involved in IRB.

2) Though the amount and dilutions of p72-MPs and p72-AE were mentioned, the amount of swine serum that was used for each assay seems not to be mentioned anywhere in the manuscript. Please consider adding the volumes of serum used for development of this assay.

Response: Thank you for your constructive comments. In this study, all samples were detected using a one-step model or two-step model, using 20 μ L of porcine serum for each assay, as described in "Detection procedure" in line 224 of the the

current version of the manuscript.

All other comments:

Introduction

1. Line 71 "To date, many different serological assays have been established." This statement needs a better phrasing to make it sound logical

Response: Thank you for your suggestion. We have included the sentences "To ensure early diagnosis of ASF, many types of serological tests have been developed." in lines 71-72 of the current version of the manuscript. If there's anything else that needs fixing, please let us know! Your comments are greatly appreciated.

2. Line 74 Expand QDM in QDM-based ASFV immunosensor

Response: Thank you, the "QDM" was quoted from a published article (Rapid and ultra-sensitive detection of African swine fever virus antibody on site using QDM based-ASFV immunosensor (QAIS), *Anal Chim Acta* . 2022 Jan 2:1189:339187). In this paper, Jiahao Li et al. established a new method to detect ASFV antibodies, and the "QDM" refer to "quantum dot microsphere (QDM) ".

2. Line 76 been discussed. Serological testes are

Response: Thank you for your careful review and for bringing this issue to our attention. We sincerely apologize for the language problems in the original manuscript. And we have corrected the spelling error.

3. Line 78-79 Among these, ELISA as most commonly used method, its time-consuming and labor..

Response: Thank you very much for your careful review. We have realized that the sentence is confusing and have revised it. We included the sentences "Of these, ELISA is the most commonly used method. However, it is time-consuming and labour-intensive and has a narrow dynamic range." in lines 78-80 of the current version of the manuscript. We hope this is now clearer.

Results

1. The CVs of inter-batch and inter-batch were tested" should be intra and inter reagents including p72-MBs = should be p72-MPs

Response: Thank you for your careful review. We have corrected "inter-batch and inter-batch" to "intra-batch and inter-batch", "p72-MBs" to "p72-MPs" in the current version of the manuscript.

2. The Figure 4 sections has E where as the legend has G, without E and F, please clarify

Response: We are very sorry for our inadvertent mistake. The legend in the original manuscript contains only ABCDE. Due to our mistake, E was written as G. This has now been corrected.

Discussion

1. The DA_gS-aCLIA is capable of detecting IgM, IgA and IgG- should be clarified in which infection/detection situations to the appropriate references 44,45

Response: Thank you for your kind comment. In our study, the DA_gS-aCLIA was established based on the principle of dual antigen sandwich. The dual antigen sandwich format, which uses a labeled antigen and a capture antigen to sandwich the target antibody, has the advantage of detecting total rather than class-specific antibodies. In particular, the total screening of all types of antibodies including IgA, IgM, IgG, etc, is advantageous for improving the detection rate of HCV. Therefore, the dual antigen sandwich based ELISA has distinct methodological advantages over indirect ELISA. It is recommended for the diagnosis of HCV infection (References 44, 45). In addition, the dual antigen lateral flow immunoassay has been used to detect total antibodies to SARS-CoV-2 (Ref. A). In this study, the dual antigen CLIA was established for the first time to detect total antibodies to ASFV. Based on these studies, we believe that the dual antigen CLIA, capable of detecting total antibody to ASFV, provides the methodological basis for improved sensitivity.

Taken together, the indirect format uses immobilized antigen and anti-pig IgG/IgM antibody to sandwich the analyte antibody. The dual antigen sandwich format can detect total antibody in all infection/detection situations.

Ref. A: Quantum dots assembly enhanced and dual-antigen sandwich structured lateral flow immunoassay of SARS-CoV-2 antibody with simultaneously high sensitivity and specificity. *Biosensors and Bioelectronics*, 2022

2. It's not clear in the below statement what the exact differentiation factor is for tosyl MPs and how they are done, ex: sample cup or another reaction chamber vs 96-well plate " Second, the advanced DAgS-aCLIA used Tosyl MPs instead of a traditional 96-well plate" Prospects and conclusions The ending "also has very good prospects for application." Should have a conclusive point instead of just application.

Response: Thank you very much for your question and comment. According to the solid phase, CLIA can be divided into tube-based format and plate-based format. Plate-based CLIA uses a 96-well or 384-well plate as the solid phase to immobilize antigen and detects antibodies by CL reaction. In contrast, the tube-based CLIA uses magnetic particles (MPs) as the solid phase to immobilize antigen, with the tube (chamber or cup) used only as a reaction vessel. In our study, a tube-based CLIA was established and had significant advantages of high throughput and automation when compared to a 96-well plate-based CLIA reported in a previous study.

Three different commercially available MPs with carboxyl groups, tosyl groups and streptavidin on their surface were used in the CLIA. Unlike previous studies, our CLIA used tosyl MPs rather than carboxyl MPs to coat antigens to detect ASFV antibodies. This is because tosyl MPs can be easily manipulated compared to carboxyl MPs, as discussed in lines 487-491 of the current version of the manuscript.

We really appreciate your comment. We have made revision and included the sentences "In conclusion, the developed DAgS-aCLIA has the advantage of high sensitivity, automation and quantitative detection of p72-Ab, which can provide important technological support for epidemiological surveillance and vaccine

development. " in lines 539-541 of the current version of the manuscript.

Re: Spectrum00809-24R1 (Development of a fully automated chemiluminescent immunoassay for the quantitative and qualitative detection of antibodies against African swine fever virus p72)

Dear Dr. Lei Wang:

Thank you for the privilege of reviewing your work. Below you will find my comments, instructions from the Spectrum editorial office, and the reviewer comments.

Revision Guidelines

Sincerely,
Vera Tesic
Editor
Microbiology Spectrum

Reviewer #1 (Comments for the Author):

The authors have made improvements to their manuscript and appropriately addressed all specific comments and questions brought to their attention by both reviewers. This manuscript presents the development of a novel automated double antigen sandwich chemiluminescent immunoassay for the detection of antibodies against ASFV p72. Thank you for the opportunity to review this manuscript.

There are still numerous, albeit mostly minor, grammatical errors that should be corrected prior to publication. While these errors do not significantly detract from the overall message or technical merit of the manuscript, further careful proofreading is recommended. Some examples (in reference to marked up version) in the first few pages are listed below.

Page 2, line 39: change "trance" to "trace"; change "rapid" to "rapidly"

Page 3, line 83: replace "and" between "time-consuming" and labour-intensive" with a comma

Page 3, line 84: remove "As"

Page 3, line 85: remove "And"

Page 4, line 105: add space between "immunoassay" and "(CLEIA)"

Page 6, line 158: change "Then" to "then"

Additionally, on page 11: replace period after "quality" with comma (line 310) and replace "MM" (line 311) with "MW". I suggest replacing "TEM revealed the nanoparticle size and morphology of p72 with a diameter of 8.46{plus minus}0.69 nm, is consistent with published data (Fig. 2C)." with "TEM revealed the nanoparticle size (with diameter of 8.46{plus minus}0.69 nm) and morphology of p72 are consistent with published data (Fig. 2C)." (lines 316-317).

On page 12, line 335: replace period with comma. Please consistently use either "labeled" or "labelled" (e.g. page 2, line 38 or page 12, line 337 versus page 12, line 336). "There are also some considerations that we would also like to mention." on page 19, line 536 can be removed entirely, as well as "one" on line 544 and the extra period on line 553.

Reviewer #2 (Comments for the Author):

Thank you for addressing all the comments. Just one minor comment on this below lines. You mentioned three and only two were described, either change text to two or add another one if any or clarify that ?

"235 Quantitative analysis. Three important parameters were determined to conduct
236 quantitative analysis"

We would like to thank the editor and reviewers for taking the time and effort to review our manuscript. We sincerely appreciate all valuable comments, which helped us to improve the quality of the manuscript. All of us have carefully studied these comments and made revisions that we hope will meet with your approval.

Reviewer #1 (Comments for the Author):

The authors have made improvements to their manuscript and appropriately addressed all specific comments and questions brought to their attention by both reviewers. This manuscript presents the development of a novel automated double antigen sandwich chemiluminescent immunoassay for the detection of antibodies against ASFV p72. Thank you for the opportunity to review this manuscript.

There are still numerous, albeit mostly minor, grammatical errors that should be corrected prior to publication. While these errors do not significantly detract from the overall message or technical merit of the manuscript, further careful proofreading is recommended. Some examples (in reference to marked up version) in the first few pages are listed below.

1. Page 2, line 39: change "trance" to "trace"; change "rapid" to "rapidly"

Response: Thank you for your careful review and kind suggestions. We sincerely apologize for the language problems in the manuscript. And we have corrected "trance" to "trace" and "rapid" to "rapidly" in line 39 of the current version of the manuscript.

2. Page 3, line 83: replace "and" between "time-consuming" and labour-intensive" with a comma

Response: Thank you for your kind suggestion. We have replaced "and" with a comma in line 79 (Page 3, line 83) of the current version of the manuscript.

3. *Page 3, line 84: remove "As"*

Response: We have removed "As" in line 80 (Page 3, line 84) of the current version of the manuscript.

4. *Page 3, line 85: remove "And"*

Response: We are grateful for the suggestion! We have removed "And" and replaced "quantification" with "Quantification " after "And" in line 81 (Page 3, line 85) of the current version of the manuscript.

5. *Page 4, line 105: add space between "immunoassay" and "(CLEIA)"*

Response: Thank you for your suggestion. We have added space between "immunoassay" and "(CLEIA)" in the manuscript.

6. *Page 6, line 158: change "Then" to "then"*

Response: We are very sorry for our careless mistake. We have corrected this error in line 154 (Page 6, line 158) of the current version of the manuscript.

7. *Additionally, on page 11: replace period after "quality" with comma (line 310) and replace "MM" (line 311) with "MW". I suggest replacing "TEM revealed the nanoparticle size and morphology of p72 with a diameter of 8.46{plus minus}0.69 nm, is consistent with published data (Fig. 2C)." with "TEM revealed the nanoparticle size (with diameter of 8.46{plus minus}0.69 nm) and morphology of p72 are consistent with published data (Fig. 2C)." (lines 316-317).*

Response: We are grateful for the suggestions! We have replaced period after "quality" with comma and replaced "MM" with "MW". We also replaced "TEM revealed the nanoparticle size and morphology of p72 with a diameter of 8.46{plus minus}0.69 nm, is consistent with published data (Fig. 2C)." with "TEM revealed the nanoparticle size (with diameter of 8.46{plus minus}0.69 nm) and morphology of p72 are consistent with published data (Fig. 2C)." in the current version of the manuscript.

8. *On page 12, line 335: replace period with comma. Please consistently use either "labeled" or "labelled" (e.g. page 2, line 38 or page 12, line 337 versus page 12, line 336). "There are also some considerations that we would also like to mention." on page 19, line 536 can be removed entirely, as well as "one" on line 544 and the extra period on line 553.*

Response: We sincerely thank you for careful review. We have replaced period with comma, and replaced "The" with "the" after comma in line 328 (page 12, line 335) of the current version of the manuscript. We consistently use "labeled" in accordance with your kind suggestion in lines 38, 98, 112, 176, 754 and 756 of the current version of the manuscript. In addition, "labeled" has been used in line 329 (page 12, line 335), while "labeling efficiency" has been used in line 330 (page 12, line 336).

Following your valuable perspective, we have removed "There are also some considerations that we would also like to mention." as well as "one" and the extra period in the current version of the manuscript.

Reviewer #2 (Comments for the Author):

1. Thank you for addressing all the comments. Just one minor comment on this below lines. You mentioned three and only two were described, either change text to two or add another one if any or clarify that ? "235 Quantitative analysis. Three important parameters were determined to conduct 236 quantitative analysis"

Response: Thank you for your questions and suggestions. "Calibration curves", "linearity range" and "lower detection limit (LDL)" have been mentioned as three important parameters affecting quantitative analysis. As "calibration curves" and "linearity range" are inextricably linked, we have described these two parts together in the original manuscript. If there's anything else that needs to be corrected, please let us know! Your comments will be greatly appreciated.

Re: Spectrum00809-24R2 (Development of a fully automated chemiluminescent immunoassay for the quantitative and qualitative detection of antibodies against African swine fever virus p72)

Dear Dr. Lei Wang:

Your manuscript has been accepted, and I am forwarding it to the ASM production staff for publication. Your paper will first be checked to make sure all elements meet the technical requirements. ASM staff will contact you if anything needs to be revised before copyediting and production can begin. Otherwise, you will be notified when your proofs are ready to be viewed.

Sincerely,
Vera Tesic
Editor
Microbiology Spectrum